# Learning to love diligent trolls: Accounting for rater effects in the dialogue safety task

**Michael John Ilagan**
McGill University / Montreal, QC, Canada
michael.ilagan@mail.mcgill.ca

## Abstract

Chatbots have the risk of generating offensive utterances, which must be avoided. Post-deployment, one way for a chatbot to continuously improve is to source utterance/label pairs from feedback by live users. However, among users are trolls, who provide training examples with incorrect labels. To de-troll training data, previous work removed training examples that have high user-aggregated cross-validation (CV) error. However, CV is expensive; and in a coordinated attack, CV may be overwhelmed by trolls in number and in consistency among themselves. In the present work, I address both limitations by proposing a solution inspired by methodology in automated essay scoring (AES): have multiple users rate each utterance, then perform latent class analysis (LCA) to infer correct labels. As it does not require GPU computations, LCA is inexpensive. In experiments, I found that the AES-like solution can infer training labels with high accuracy when trolls are consistent, even when trolls are the majority.

## 1 Introduction

For chatbots, it is not enough to learn to generate coherent utterances—many coherent utterances are undesirable, such as offensive remarks. Post-deployment, to have the chatbot continuously improve its judgment, one way is to source training examples from feedback by live organic users (Shuster et al., 2022; Xu et al., 2022). Though more realistic than crowdworkers, among organic users are trolls, whose erroneous feedback fosters bad behavior (Wolf et al., 2017). How can the chatbot learn continuously from user interactions in a way that is robust to trolls?

The problem can be cast as a dialogue safety task (Dinan et al., 2019; Noever, 2018): Given input utterance $x$, output its binary class label $y$, either safe (denoted $y = 0$) or unsafe (denoted $y = 1$). The training set is then a set of observed $(x, y^*)$

| | Grader 1 | Grader 2 | Grader 3 |
|---|---|---|---|
| Essay 1 | 1 | | 0 |
| Essay 2 | | 1 | 0 |
| Essay 3 | 0 | 0 | |
| Essay 4 | 0 | | 1 |
| Essay 5 | | 0 | 1 |

Table 1: An essay $\times$ grader inter-rater matrix (denoted $Z$ in the text) of annotations. Here, the grades are binary (e.g. pass/fail). Each essay is assigned to a random subset of the graders. Rater effects must be accounted for, so that a final score can be assigned to each essay for training an automated essay scoring (AES) engine.

utterance/label pairs sourced from users. But due to trolls (as well as honest mistakes), some of the training pairs have incorrect labels (i.e. $y^* \neq y$). We thus endeavor to de-troll these labels, as a pre-processing step (Song et al., 2022). Once labels are corrected (i.e. $y^* = y$), the problem reduces to familiar supervised learning.

Toward de-trolling the training set, a two-intervention solution was recently offered by Ju et al. (2022). **Intervention 1** has each user provide multiple $(x, y^*)$ pairs, so examples are nested within users. **Intervention 2** computes the "untrustworthiness" of each training example via (3-fold) cross-validation (CV). A training example is removed if it exceeds some threshold of disagreement with the crowd. The intuition is that trolls end up having a bad track record against the crowd but occasionally, by chance, provide examples worth keeping. The $(x, y^*)$ pairs remaining are then suitable for training.[1]

There are two issues with this CV-based (Ju et al., 2022) solution. First, as it involves CV on large language model (LLM) computations, it is expensive. Second, CV relies upon an abundance of *helpers* (Ju et al., 2022), good-faith users who end up in

---

[1]Alternatively, training pairs exceeding the threshold are kept but with their labels flipped (Ju et al., 2022).

agreement with each other. Thus, this approach can be overwhelmed in the event of a coordinated attack where trolls have both strength in numbers and are consistent among themselves.

A promise of circumventing both issues lies in methodology for automated essay scoring (AES) (Attali and Burstein, 2006; Taghipour and Ng, 2016; Uto, 2021). Given an input essay $x$, an AES engine (which may incorporate an LLM) must output a score $y$, a numeric or ordinal rating of the essay's quality. The burden of annotating a training set is divided among multiple human graders, which incurs rater effects—graders vary on how strict or permissive they are, which adds variance to automated scoring (Wind et al., 2018; Uto and Okano, 2020).

Analogously, the AES solution has two interventions (Uto and Okano, 2020). **Intervention 1** induces redundancy by randomly assigning multiple graders (users) to each training essay (utterance), yielding an inter-rater matrix $Z$ as in Table 1. **Intervention 2** fits a rater effects model to $Z$, a statistical (as opposed to neural) model, which yields inference, for each unique essay (in dialogue safety: utterance), a single final score (in dialogue safety: class label) $\tilde{y}$ suitable for training. Such a solution is inexpensive, being free of embeddings— the essays' content $x$ is not incorporated in $Z$ (as in Table 1) and is not needed to infer $\tilde{y}$. In the case of discrete ratings, the rater effects model is called a latent class analysis (LCA) model (Linzer and Lewis, 2011).

In the present work, my contribution is twofold. First, I establish a connection between the chatbot robust learning (e.g. Dinan et al., 2019) and the AES (e.g. Attali and Burstein, 2006) literatures. Second, I conduct an examination of an AES-like solution to de-trolling the training set. The main result is that this solution thrives on consistency, even among trolls: when trolls are consistent among themselves and are majority of the users, inferred labels are accurate; but curiously, when trolls are majority but make only random noise, the AES-like solution is not as accurate.

## 2   Related work

In open dialogue models, the capacity to improve past pre-training is desired. Dinan et al. (2019) found that challenging crowdworkers to "break" a pre-trained classifier with adversarial inputs in follow-up rounds of training was helpful toward making the classifier robust. Xu et al. (2022) considered various methods to improve chatbots in deployment from different forms of human feedback, including free-form text. An emerging innovation is architectures that are self-monitoring, being both generator and classifier (Arora et al., 2022; Lu et al., 2022). But as such architectures assume that training examples have correct labels, they are solutions for supervised learning, not detrolling. Addressing the problem takes more than finding a state-of-the-art classifier.

The threat posed by trolls falls under the umbrella of robust learning (Song et al., 2022). Whereas the CV-based procedure (Ju et al., 2022) is one way to do robust learning, another is expectation-maximization (EM) algorithms (Do and Batzoglou, 2008; Boos and Stefanski, 2013), of which LCA fitting (Linzer and Lewis, 2011) is an instance. In statistics, EM is a local optimization technique for fitting models involving latent variables, such as unobserved class labels. An elementary instance of EM is the familiar $k$-means clustering algorithm (Do and Batzoglou, 2008). More sophisticated instances of EM are applicable to problems with untrustworthy training examples and rater effects, such as crowdworkers' tagging of images and documents (Ipeirotis et al., 2010; Raykar and Yu, 2012).

The use of AES in large-scale educational assessment dates back decades (Attali and Burstein, 2006). The AES literature goes deeper—as essays may have been written in response to different prompts, some not appearing in training, and some assessments score essays in multiple criteria (Uto, 2021). But in the present work, the most basic AES setup suffices for the analogy—every annotation $y^*$ (the class label or score) is a commonsense holistic evaluation of the safety or toxicity of $x$ (the utterance or essay).

While the present article is focused on toxicity or safety (Dinan et al., 2019; Noever, 2018), there are other desiderata for chatbot behavior, such as factual correctness and staying on-topic (Xu et al., 2022; Arora et al., 2022).

## 3   From ratings to predicted classes

To motivate LCA, consider Table 1, an intuitive case. By inspection, Graders 1 and 2 are in agreement, and Grader 3 goes in the opposite direction.[2]

---

[2]I am not suggesting that educational assessments risk trolling, as they are not crowdsourced. Non-troll rater effects

Figure 1: A latent class analysis (LCA) model for dialogue safety. Suppose Cluster A is the safe cluster. Then User 1 must be a helper, as they have a 91% probability of labeling a safe utterance as such, and they have an 86% probability of labeling an unsafe utterance as such. From similar reasoning, User 2 must be a troll. But if Cluster A were unsafe instead, the roles are the opposite.

Accordingly, there are two clusters: those rated "0" by Grader 3 (i.e. Essays 1 and 2); and those rated "0" by Graders 1 and 2 (i.e. Essays 3 to 5). Realistic instances of $Z$ are not as clear-cut, so it takes a statistical model to apply such intuitions at scale.

The LCA model is depicted in Figure 1. Despite the name, LCA deals with clusters rather than classes. According to this model, each utterance belongs to one of two clusters, A and B, with a probability distribution (the *prior*). Within each cluster, the distribution of the observed label (the *likelihood*) is user-specific: each user has some probability of labeling a Cluster A utterance as safe, and some probability of labeling a Cluster B utterance as safe.[3]

An EM algorithm (Linzer and Lewis, 2011), provided $Z$, estimates all parameters (in both prior and likelihood). The intuition is that users' tendencies can be quantified due to utterances shared between them. Provided these estimates, for each utterance, the *posterior* distribution can be computed, which is the answer to the question, "Given the observed pattern of labels (i.e. row of $Z$), which cluster is more probable?" Each utterance is then assigned to the highest-probability cluster. Note that LCA fitting is possible even amid missing values in $Z$, but some redundancy (from Intervention 1) is required (see Limitations section for details).

However, clusters are not yet classes. In the dialogue safety task, trolls and helpers behave in

opposite directions, and which is the "correct direction" cannot be inferred from merely looking at $Z$, as in Table 1. To be able to map clusters to classes in the present work, I make the assumption that the cluster having more utterances is the safe class. Such an assumption is true of the dataset (Dinan et al., 2019) used in experiments in the next section. Furthermore, if the training examples are feedback on the chatbot's own utterances (Xu et al., 2022), then any decent chatbot should produce safe examples more than half the time.

Altogether, the AES-like solution consists of (Intervention 1) introducing redundancy, then (Intervention 2) statistically leveraging it via LCA with the safe-as-majority assumption (LCA+SM). The end result is corrected training pairs $(x, \tilde{y})$. The de-trolling solution is inexpensive because it is embeddings-free; and it is robust because user tendencies are accounted for, without taking for granted that the consensus is correct. Unlike the CV-based solution (Ju et al., 2022), no untrustworthiness scores need be calculated, as user tendencies are already accounted for. No examples are removed from training—the intuition is that a user that reliably gives incorrect answers is perfectly good data once their labels are flipped.

## 4 Experiments

### 4.1 Methods

Experiments were conducted to test the efficiency and robustness of the AES-like solution under different troll scenarios, especially extreme cases of trolls being majority not covered in Ju et al. (2022). Real utterances were labeled by synthetic users. The real utterances $x$, with gold labels $y$ known, were from Meta AI's single-turn safety dataset (Dinan et al., 2019). Unfortunately due to computational resource constraints, there was no direct comparison to the CV-based solution in Ju et al. (2022).[4] Instead, the proposed Intervention 2, LCA+SM, was compared to a baseline of majority vote (MV) on each row of $Z$. Both solutions leverage redundancy, though MV naively takes for granted that the consensus is correct; and both directly impute classes rather than rank examples by untrustworthiness.

Two Experiments followed the same $2 \times 2 \times 2 \times 2$

---

exist, and Table 1 resembling trolling is simply to serve the analogy to the dialogue safety task.

[3]Alternatively, the likelihood can be expressed in terms of user-specific probability transition matrices, as in Ju et al. (2022).

[4]CV-based trustworthiness scores are a function of two hyperparameters (one of them is the threshold) that themselves require tuning with a validation set. Thus, there are two layers of validation.

design, varying the following factors:

UNSAFE_PREVALENCE: Either **10%** or **30%** of the utterances were unsafe, in both training and validation sets.

TROLL_PREVALENCE: Either **50%** or **90%** of the users were trolls.

CORRUPT_ACTION: When corruption of the label occurred, it was either "**diligent**" ($y^* = 1 - y$) or "**lazy**" (randomly choose between 0 and 1 with equal probability).

TROLL_CORRUPT_RATE: Each troll corrupted either **80%** or **95%** of their labels.

**Experiment 1** had both a de-trolling phase and a supervised learning phase.

1. **De-trolling.** 200 utterances were sampled as training set, and $Z$ was created by randomly assigning each utterance to 5 users out of a pool of 50 users. Thus, $Z$ had 90% of its cells missing. To incorporate honest mistakes, helpers corrupted 5% of their labels in all conditions. Both competing methods (i.e. LCA+SM and MV) were applied to yield $(x, \tilde{y})$ pairs.

2. **Supervised learning.** 24 utterances were sampled as validation set, with gold labels known for simplicity. A classifier was trained on the corrected training set, with the validation set used for early stopping. Constant across all runs, the test set was the same as in Ju et al. (2022), which had 900 safe utterances and 100 unsafe utterances.

Accordingly, the evaluation metric in Experiment 1 was test set accuracy. Due to the computational expense, there were only 5 runs per scenario.

To fit LCA, I used the package `mirt` in R. For training, I used ParlAI (Miller et al., 2017), using the same settings as in Ju et al. (2022) except for omitting CV and setting `max-train-steps=400` to keep the experiment manageable. The neural model, called `bi_model_huge_reddit` in ParlAI's Model Zoo, consisted of a linear layer on top of a pre-trained transformer from Humeau et al. (2019).

While evaluating downstream accuracy is usually sought, an issue in Experiment 1 is that such a metric conflates the de-trolling solution with the

supervised learning solution. For instance, bad performance might be blamed on a poor neural classifier or on utterances being intrinsically difficult, rather than the merits of the de-trolling solution itself. Thus, **Experiment 2** redid the de-trolling phase with the same settings except that the metric was instead the accuracy of the imputed labels $\tilde{y}$ in the training set—I call this metric *imputation accuracy*.[5] The test set is thus irrelevant. Unburdened by the expense of training a large language model, each scenario had 500 runs.

## 4.2 Results

With GPU for the supervised learning phase, Experiment 1 ran for ~12.1 hours. In contrast, without need for GPU, Experiment 2 ran for ~2.2 hours.

Figure 2 shows the results for both Experiments. Only results for UNSAFE_PREVALENCE=30% are shown. The patterns for UNSAFE_PREVALENCE=10% are similar and can be found in the Appendix.

Trends were similar between the two Experiments, so I describe them together. Keep in mind that when CORRUPT_ACTION="diligent", a higher TROLL_CORRUPT_RATE means that trolls are more consistent among themselves, as they are strongly negatively correlated with ground truth; but when CORRUPT_ACTION="lazy", a higher TROLL_CORRUPT_RATE means that trolls are less consistent among themselves, due to their randomness. The overall picture is that LCA+SM leverages consistency in the user-provided labels, even malicious consistency from trolls; but LCA+SM falters when there is little consistency to leverage. Under CORRUPT_ACTION="diligent", LCA+SM was highly accurate when TROLL_CORRUPT_RATE=95% and TROLL_PREVALENCE=90%, always better than the baseline; but under CORRUPT_ACTION="lazy", LCA+SM was highly inaccurate for the same combination, sometimes worse than the baseline. Malicious as trolls are, the diligent ones are just as valuable as helpers—they should not be removed from training.

## 5 Discussion

How much data should $Z$ have for LCA to be reliable? While the proportion of missing values was

---

[5] The connotations of overfitting and over-optimism usually associated with "training set accuracy" are not applicable here. Yes, both LCA+SM and MV are used on the training set; but both are unsupervised and do not overfit.

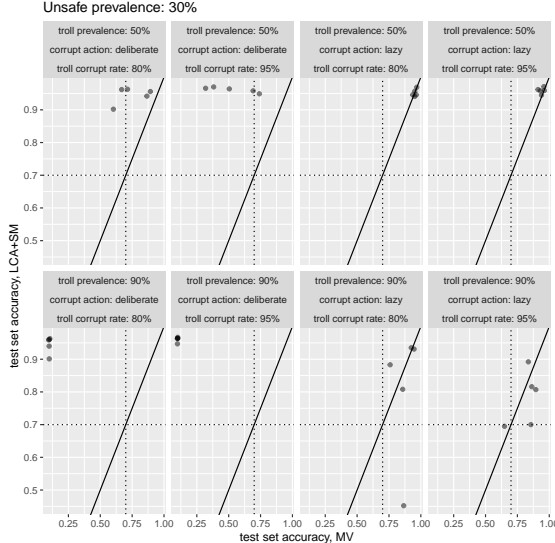

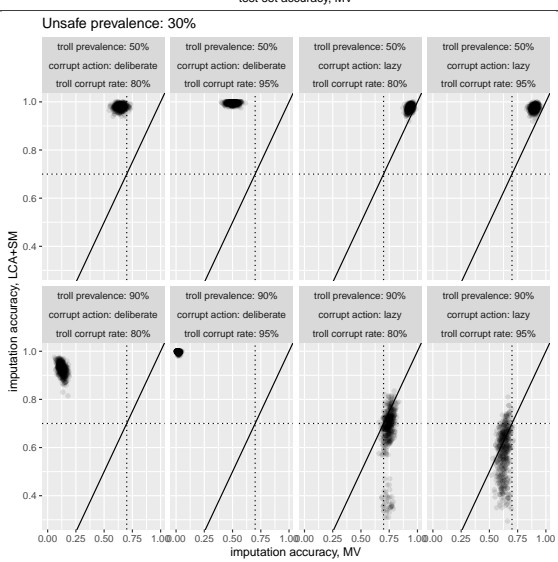

Figure 2: Results for both Experiments, for 30% unsafe prevalence. Within each Experiment, each panel is a scenario. Solid black line is the identity line—when a point is above the line, LCA+SM did better than MV. Dotted lines mark the accuracy of indiscriminately predicting safe. LCA+SM = latent class analysis with safe-as-majority assumption; MV = majority vote.

not manipulated in the Experiments, exploratory analysis found that larger non-missing proportions were more favorable: increasing the users assigned per utterance was beneficial for fixed total user pool; but for fixed number of users per utterance, increasing the user pool was detrimental. While having more data is statistically expedient, it also imposes a greater burden on the user (Shuster et al., 2022). In these Experiments, having 10% of the inter-rater matrix filled meant that each user labeled on average 20 utterances, which is 5x what the CV-based approach (Ju et al., 2022) would have

required. Note that having a very low non-missing proportion may make fitting LCA impossible.

While the CV-based (Ju et al., 2022) solution's issues were not directly experimentally validated, there is reason to believe that it would have been at a disadvantage against the AES-like solution. In terms of expense, CV would involve training a large language model, even at the de-trolling phase. In terms of robustness, CV can be thought of as simulating "checking each other's work" or tapping into a consensus in a way that MV directly does, so the CV-based solution would have been likewise vulnerable to a false consensus. Scenarios where the AES-like solution faltered had little consensus, which would have been unfavorable to the CV-based solution as well.

Unlike Ju et al. (2022), my Experiments did not manipulate the prevalence of adversarial vs. non-adversarial utterances sampled. In the dataset (Dinan et al., 2019), half of the utterances were adversarial. These prevalences would have affected the supervised learning phase but not the de-trolling phase, as MV and LCA+SM are embeddings-free.

It turns out that Experiment 2 was a purely statistical exercise, though one relevant to the de-trolling problem. After all, producing $Z$ and evaluating imputation accuracy did not reference any utterance content, so no real data was needed. Consequently, the results for imputation accuracy are widely generalizable in the sense that they are irrespective of the choice of neural classifier or the intrinsic difficulty of the utterances being classified. In addition, Experiment 2 can be expediently adapted to different settings (e.g. UNSAFE_PREVALENCE and number of training utterances) for researchers to mess around with. But for specific applications involving a specific dataset and neural classifier, downstream accuracy becomes of interest.

# 6   Conclusion

The AES-like solution is inexpensive, as it does not require GPU computation.

In the event of a coordinated attack where trolls are majority and consistent among themselves, the AES-like solution holds up well, bearing the wisdom that "diligent" trolls may be leveraged toward better performance. But when there is little consensus among users, there is little to work with, so de-trolling accuracy drops.

## Limitations

The AES-like solution proposed inherently has higher data requirements, which imposes a burden on the users. Logistical and privacy concerns may be raised in light of having users "check each other's work", especially when example utterances are from users' own conversations with the chatbot.

To map clusters to classes, the present work assumed SM. MV and the CV-based (Ju et al., 2022) solution put trust in good-faith behavior of users, whereas LCA+SM gains robustness by putting trust in the chatbot's decency instead. Either way, trust has to be put somewhere. That said, even if SM were true more generally, it could still fail to hold for some batches in continual learning, if the batches are small enough.

Not every inter-rater matrix $Z$ can be fitted to LCA. To make fitting possible, there are two requirements (Chalmers, 2012).

(1) As each column (i.e. user) is associated with two parameters, the number of rows (i.e. utterances) in must be at least twice the columns.

(2) Each column must have both possible labels.

If (2) is violated by a column, that column may be deleted before fitting, as long as (1) is still met. Grave scenarios of trolling may yield many invalid columns, deletion of which would make fitting LCA impossible.

## Acknowledgements

Inspiration for the present topic came from a conversation with Nicholas Meade. The use of LCA came from a suggestion by Carl Falk.

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

## A   Appendix

Figure 3 shows the results for UN-SAFE_PREVALENCE=10%.

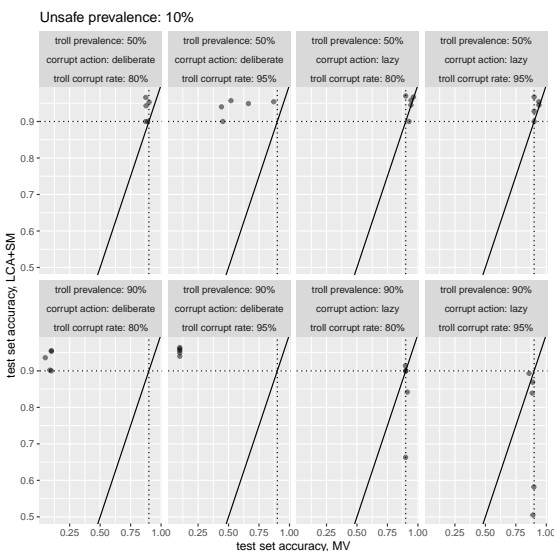

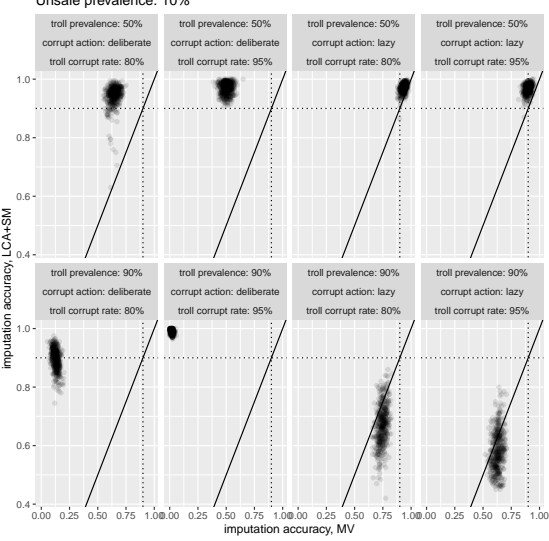

Figure 3: Results for both Experiments, for 10% unsafe prevalence. Within each Experiment, each panel is a scenario. Solid black line is the identity line—when a point is above the line, LCA+SM did better than MV. Dotted lines mark the accuracy of indiscriminately predicting safe. LCA+SM = latent class analysis with safe-as-majority assumption; MV = majority vote.