# OpenReview forum: "Learning to love diligent trolls: Accounting for rater effects in the dialogue safety task"
_EMNLP/2023/Conference — EMNLP 2023 Findings_

### Official Review · Reviewer_UqMy · 2023-08-05

**Soundness:** 4

**Excitement:**

4: Strong: This paper deepens the understanding of some phenomenon or lowers the barriers to an existing research direction.

**Paper Topic And Main Contributions:**

This paper delves into the challenges presented by adversarial noise, specifically from "trolls", in corrupting data for machine learning models. Traditionally, training examples with high user-aggregated cross-validation (CV) error were omitted to counteract this "trolling". However, CV, being computationally expensive, can be rendered ineffective in situations of coordinated troll attacks. Addressing this, the paper introduces a novel solution inspired by automated essay scoring (AES) methods: garner ratings from multiple users for each utterance and subsequently employ latent class analysis (LCA) to deduce accurate labels. Notably, LCA, avoiding GPU-intensive computations, proves to be cost-effective.  Through experimental validation, it was demonstrated that this AES-inspired approach is adept at inferring correct training labels, even when the majority of feedback originates from coordinated trolls. The research concludes with an intriguing observation: when harnessed effectively, even "diligent" trolls can inadvertently improve classifier performance.

**Reasons To Accept:**

- Excellent writing style, concise language and clear logic.
- The issue of chatbots inadvertently generating offensive utterances is crucial. Improving the quality and safety of chatbot responses is pertinent to both industry and society.
- The paper offers a novel solution inspired by automated essay scoring (AES) methodologies. The introduced latent class analysis (LCA) method, in contrast to GPU-intensive computations, is highlighted as an inexpensive solution, making it more scalable and feasible for real-world applications.

**Reasons To Reject:**

In this paper, the authors repeatedly emphasize the simplicity and cheapness of this method, but in the experiments, they did not compare it with Ju et al 2022 mentioned in the paper, but only with Dummy Classifier (MV), which is a weak comparison. Although the results of the method in this paper may not be as good as Ju et al 2022, I think this method is a very good comparison if it is not too much worse.

**Reproducibility:**

4: Could mostly reproduce the results, but there may be some variation because of sample variance or minor variations in their interpretation of the protocol or method.

**Reviewer Confidence:**

2: Willing to defend my evaluation, but it is fairly likely that I missed some details, didn't understand some central points, or can't be sure about the novelty of the work.

---

> ### Author Rebuttal · Authors · 2023-08-29
>
> This review was absolutely a delight to read.
> The Reviewer's summary of the present paper was brilliant.
> I had to stop and check that I wasn't mistakenly reading a paragraph I wrote myself.
>
> The Reviewer said:
>
> > in the experiments, they did not compare it with Ju et al 2022 mentioned in the paper, but only with Dummy Classifer (MV), which is a weak comparison. Although the results of the method in this paper may not be as good as Ju et al 2022, I think this method is a very good comparison if it is not too much worse.
>
> I confirm the Reviewer's claim that the comparison was to MV (majority vote) and not exactly the CV (cross-validation)-based method in Ju et al (2022).
> Let me explain.
>
> # Why not the CV-based method?
>
> Let's walk through Ju et al's (2022) method, with a little more detail.
>
> 1. **(Intervention 1)** Use 3-fold CV with an LLM.
> The result is CV-predicted unsafe probabilities per utterance.
> Note that doing so simulates "checking each other's work" or tapping into some wisdom of the crowd.
> 2. **(Intervention 2)** Filter the training data by untrustworthiness scores.
> The formulas are in Ju et al (2022), but the idea is that labels far from the CV predictions are less trustworthy.
> Apply a threshold for untrustworthiness, a hyperparameter $\beta$.
> The result is a training set with fewer examples, retaining their original user labels.
> (It mentions the possibility of flipping the labels below the threshold instead, but they went for deletion in their experiments.)
>
> Unfortunately, Ju et al's (2022) full procedure was prohibitively difficult to reproduce.
> Their Github code shows only up to how to get the CV predictions (Intervention 1) in ParlAI.
> But the rest (Intervention 2) is just described in the text with some formulas, no ParlAI code, and the grid to be searched for $\beta$ was not given.
> Note that choosing $\beta$ using a validation set (on top of the CV predictions), as described in Ju et al (2022), would be expensive.
> Imagine fine-tuning the LLM with all training examples, then again dropping the most untrustworthy example, then dropping the two most untrustworthy examples, and so on.
>
> I want to caution the Reviewer about comparing results to those reported in Ju et al (2022).
> Ju et al's (2022) experiments did not have a condition where trolls were a majority and consistent with each other, which is precisely what motivated the present paper.
>
> # Why majority vote?
>
> Now let's recap my proposed AES (automated essay scoring)-like solution.
>
> 1. **(Intervention 1)** Assign multiple users per utterance.
> The result is the $Z$ matrix.
> Note that doing so, rather than simulating asking the crowd, actually takes redundant labels, as in AES.
> 2. **(Intervention 2)** Apply LCA (latent class analysis).
> Each utterance is assigned a cluster, and the bigger cluster is labeled the safe class.
> The result is a final imputed label per utterance.
>
> LCA cannot be done without Intervention 1.
> But with Intervention 1, one might contend that it is just carried by the additional data collected.
> Thus, I chose a baseline that would also benefit from Intervention 1.
> The most obvious to me was MV (majority vote) (Ipeirotis et al 2010).
> Note that MV would also be vulnerable to a false consensus, like Ju et al's (2022) method, just with actual redundant labels rather than simulated consensus.
>
> # More experiments
>
> Having said all that, I still wished to satiate the Reviewer's curiosity.
> With the same settings as Experiment 2, I did new runs, for the Ju et al (2022) method, only up to producing CV predictions (i.e. Intervention 1).
> Let's call this "CV1", to emphasize that we have only one user per utterance (no redundancy, only one non-missing cell per row of $Z$).
> Then I did 50 runs of MV and 50 runs of LCA.
> Let's call these "MV5" and "LCA5" respectively, to emphasize that we have 5 users per utterance (with redundancy).
> But what metric to use?
> The metric would have to satisfy two properties.
>
> * The metric must be intrinsic (instead of downstream test set accuracy), so as to avoid expensive GPU computations.
> * The metric must be applicable to both a trustworthiness-ranking method (CV1) and a class-imputing method (MV5 and LCA5).
>
> To arrive at such a metric, I did as follows.
>
> 1. I computed untrustworthiness score equivalents for MV5 and LCA5.
> Take the squared difference between the predicted probability and the observed label.
> For MV5, the predicted probability is just the proportion of votes for unsafe (or mean of the row in $Z$).
> Note that there are as many scores as there are non-missing cells in $Z$.
> 2. For each method, I created a binary "correctness" variable: the class is "correct" if the observed label matches the gold label, and "incorrect" otherwise.
> Again, there are as many binary values as there are non-missing cells in $Z$.
> 3. For each method, the metric was the area under the ROC curve for predicting correctness from trustworthiness. Is the untrustworthiness predictive of whether the label was incorrect? Higher values are better.
>
> Scenario means are in the table below.
> For diligent-troll majority scenarios, CV1 was better than MV5, but LCA5 was better than both.
> But for lazy-troll majority scenarios, MV5 was better than the other two.
>
> |unsafe_prevalence|troll_prevalence|corrupt_action|troll_corrupt_rate|lca5_auc         |mv5_auc            |cv1_auc          |
> |-----------------|----------------|--------------|------------------|-----------------|-------------------|-----------------|
> |0.1              |0.5             |diligent      |0.8               |0.974|0.641  |0.596|
> |0.3              |0.5             |diligent      |0.8               |0.994|0.658   |0.564|
> |0.1              |0.9             |diligent      |0.8               |0.938|0.079|0.431|
> |0.3              |0.9             |diligent      |0.8               |0.945  |0.084 |0.376|
> |0.1              |0.5             |lazy          |0.8               |0.990|0.951   |0.611|
> |0.3              |0.5             |lazy          |0.8               |0.992|0.958  |0.692|
> |0.1              |0.9             |lazy          |0.8               |0.723|0.754  |0.548|
> |0.3              |0.9             |lazy          |0.8               |0.699|0.763  |0.551|
> |0.1              |0.5             |diligent      |0.95              |0.988|0.479  |0.448|
> |0.3              |0.5             |diligent      |0.95              |0.999|0.491  |0.550|
> |0.1              |0.9             |diligent      |0.95              |0.997|0.005|0.321|
> |0.3              |0.9             |diligent      |0.95              |0.999|0.006|0.422|
> |0.1              |0.5             |lazy          |0.95              |0.990|0.920  |0.610|
> |0.3              |0.5             |lazy          |0.95              |0.994|0.929  |0.707 |
> |0.1              |0.9             |lazy          |0.95              |0.603 |0.623  |0.489|
> |0.3              |0.9             |lazy          |0.95              |0.619|0.632  |0.534|
>
> ----
>
> **References**
>
> Ipeirotis, P.G., Provost, F., Wang, J. (2010). Quality management on Amazon Mechanical Turk. ACM SIGKDD workshop on human computation, 64--67. https://www.ipeirotis.com/wp-content/uploads/2012/01/hcomp2010.pdf
>
> Ju, D., Xu, J., Boureau, Y.-L., & Weston, J. (2022). Learning from data in the mixed adversarial non-adversarial case: Finding the helpers and ignoring the trolls. https://doi.org/10.48550/ARXIV.2208.03295

---

### Official Review · Reviewer_Ysnq · 2023-08-11

**Soundness:** 3

**Excitement:**

3: Ambivalent: It has merits (e.g., it reports state-of-the-art results, the idea is nice), but there are key weaknesses (e.g., it describes incremental work), and it can significantly benefit from another round of revision. However, I won't object to accepting it if my co-reviewers champion it.

**Paper Topic And Main Contributions:**

The paper addresses the challenge of "trolls" who provide maliciously incorrect feedback to chatbots, leading to the generation of inappropriate or offensive responses. The issue is framed as a dialogue safety task, wherein input utterances are classified as either safe or unsafe. The conventional approach to combat trolling, which uses cross-validation based on user-aggregated behavior, is critiqued for its computational expense and vulnerability to coordinated troll attacks. Instead, the author introduces a method inspired by automated essay scoring (AES). This involves having multiple users rate each utterance and then employing latent class analysis to infer the correct labels. The paper finds that this AES-inspired approach is effective, particularly when trolls consistently provide similar feedback, even if they are in the majority. When trolls provide random feedback, the method's accuracy diminishes.

**Questions For The Authors:**

1. How does the model scale?
2. How does the model perform compared with large language models?

**Reasons To Accept:**

The proposed LCA method does not require GPU computations, making it a more cost-effective solution compared to cross-validation.
Robustness: The AES-like solution demonstrated its effectiveness, particularly when faced with coordinated attacks by trolls. The paper establishes a connection between the challenges faced in chatbot robust learning and the methodologies employed in the AES domain.

**Reasons To Reject:**

1. The paper is not well represented. Both the writing and formatting could be improved.
2. The dataset size is extremely small (around one thousand) in crowd-sourcing context. I doubt the scalability of the proposed approach.
3. The proposed method is not compared with any basic large language model, let alone the state-of-the-art models, making the empirical advantage look in-convincing to me.
4. The efficacy of the AES-like solution is contingent on trolls being consistent in their feedback, which might not always be the case.
The method's performance drops significantly when trolls provide random feedback, indicating a potential vulnerability.

**Reproducibility:**

4: Could mostly reproduce the results, but there may be some variation because of sample variance or minor variations in their interpretation of the protocol or method.

**Reviewer Confidence:**

3: Pretty sure, but there's a chance I missed something. Although I have a good feel for this area in general, I did not carefully check the paper's details, e.g., the math, experimental design, or novelty.

---

### Official Review · Reviewer_VSeF · 2023-08-14

**Soundness:** 2

**Excitement:**

3: Ambivalent: It has merits (e.g., it reports state-of-the-art results, the idea is nice), but there are key weaknesses (e.g., it describes incremental work), and it can significantly benefit from another round of revision. However, I won't object to accepting it if my co-reviewers champion it.

**Paper Topic And Main Contributions:**

The paper considers the problem of identifying trolls, or live users who wrongly classify the model's performance (on purpose or by mistake). To this end, the paper applies a method from automated essay scoring to identify the malevolent users.  The method is an application of expectation-maximisation algorithm to the problem of troll identification.

**Questions For The Authors:**

* In the introduction the paper describes that it would be able to identify the cases where the trolls constitute a majority of the annotators. At the same time, in LL. 153--158 it is mentioned that the cluster with more utterances is assumed to be 'safe'. Could you please clarify whether such an assumption would be correct if the 'trolls' were the majority and labelled the text as unsafe?

**Reasons To Accept:**

* the paper identified an interesting problem and proposed a potential solution which leads to improved performance;
* the proposed solution is computationally light;

**Reasons To Reject:**

* the paper does not provide a comparison to any baseline model
* the method is only tested on synthetic data

**Reproducibility:**

3: Could reproduce the results with some difficulty. The settings of parameters are underspecified or subjectively determined; the training/evaluation data are not widely available.

**Reviewer Confidence:**

4: Quite sure. I tried to check the important points carefully. It's unlikely, though conceivable, that I missed something that should affect my ratings.

---

> ### Author Rebuttal · Authors · 2023-08-29
>
> Thank you for the review.
>
> # Reconciling two statements
>
> The Reviewer said:
>
> > In the introduction the paper describes that it would be able to identify the cases where the trolls constitute a majority of the annotators. At the same time, in LL. 153--158 it is mentioned that the cluster with more utterances is assumed to be 'safe'. Could you please clarify whether such an assumption would be correct if the 'trolls' were the majority and labelled the text as unsafe?
>
> Sure, let me clarify. The two statements in question are:
>
> 1. Most utterances are safe.
> 2. Most users are trolls. (So most utterances are labeled unsafe.)
>
> Yes, both statements can be true at the same time, no contradiction.
> When I said (1), I meant precisely in terms of the gold labels.
> Sorry if that was confusing.
> Of course, the users may provide observed labels that are false.
>
> Let's walk through an intuitive example.
> Suppose we have three users: Alice, Bob, and Charmaine.
> Alice and Bob completely agree with each other; but Charmaine says the opposite.
> Consequently, there'll be two clusters.
>
> * Cluster D: utterances that Alice and Bob say are safe but Charmaine says is unsafe.
> * Cluster E: utterances that Charmaine says is safe but Alice and Bob say are unsafe.
>
> Finally, suppose that Cluster E ended up larger.
> So which of the users are lying?
> If we knew the chatbot to have a basic decency so that most utterances are safe (you wouldn't deploy it otherwise), then we catch Alice and Bob (the majority) as lying.
> By similar reasoning, those lying would be caught as well if they were the minority.
>
> LCA weaponizes these intuitions formally.
> It works even if a user's labels were neither 100% true nor 100% false.
> And it works even if each utterance is labeled by only a subset of the users.
> Each utterance gets a probability of being unsafe.
>
> Also, another point of clarification.
> The Reviewer said:
>
> > the paper applies a method from automated essay scoring to identify the malevolent users
>
> I want to emphasize that the end result of the AES-like solution is not to classify users as trolls or helpers (Line 145--147).
> What it does to the users is compute their best-fitting likelihood parameters, as shown in Figure 1 (above Line 118), where essentially, each row is a user-specific confusion matrix.
> There is no cut-off for "helper" or "troll" on these parameters, and none are needed.
> Given the parameters, for each utterance, its probability of being unsafe can be computed, considering that safe is the bigger class.
>
> # Baseline model
>
> The Reviewer said:
>
> > the paper does not provide a comparison to any baseline model
>
> To set up the answer, when faced with trolls, there are two problems to solve (Lines 40--43).
>
> * **(P1)** Given input/output pairs with dubious veracity (i.e. trolled training set), produce input/output pairs that can be trusted (i.e. de-trolled training set).
> * **(P2)** Given input/output pairs that can be trusted (i.e. de-trolled training set), predict outputs for new inputs.
>
> Notice that P2 is just the familiar supervised classification task, which is not of immediate interest.
> We are not asking what classifier would be great to leverage labels that we trust.
> Instead, what we care about is the solution to P1.
> We are asking, can we arrive at labels that we can trust? Does the solution impute classes accurately?
>
> I wish the Reviewer had said more on what they were expecting as a "baseline model".
> Due to the P1-P2 sequence, there are two ways I can interpret it.
>
> * **(P1)** Did the Reviewer mean I failed to compare to a P1 baseline?
> I must disagree.
> The proposed method was to (Intervention 1) have multiple-users-per-utterance redundancy, then (Intervention 2) apply LCA.
> As a baseline, I chose (Intervention 1) having multiple-users-per-utterance redundancy, then (Intervention 2) apply MV (majority vote).
> Note that both attempt to leverage redundancy, but one does it wisely (LCA) while the other naively (MV).
> * **(P2)** Did the Reviewer mean I failed to compare to a P2 baseline?
> No comparison was intended, as there were none of immediate interest.
> To be clear, in Experiment 1, an LLM (same across scenarios) was fine-tuned after de-trolling the training data, and accuracy was computed on a test set.
> Note that this metric conflated P2 performance and P1 performance.
> Thus, I did Experiment 2, which measured the accuracy of the imputation itself in the training set.
> Otherwise, one might contend that my good results were just carried by a good pretrained classifier or an easy discrimination task.
> Note that the metric was given the name "training set accuracy" (Line 233), which is regrettable---the imputation method is unsupervised, so there is no concern that the evaluation understates error.
>
> If both of these interpretations are incorrect, I will be happy to explain if the Reviewer clarifies what "baseline" they were expecting.
>
> # Synthetic data
>
> The Reviewer said:
>
> > the method is only tested on synthetic data
>
> I must emphasize that the utterances and their gold labels are from real data in Dinan et al (2019).
> For the present paper, what I manipulated was the troll behaviors, troll prevalence, and unsafe prevalence, to see how they impact the performance of the AES-like solution.
> Monte Carlo experiments are not unprecedented, as in Howcroft & Reiser (2021).
>
> Ju et al (2022) performed an experiment on real deployment data but have not made it publicly available.
>
> ----
>
> **References:**
>
> Dinan, E., Humeau, S., Chintagunta, B., & Weston, J. (2019). Build it break it fix it for dialogue safety: Robustness from adversarial human attack. EMNLP 2019. https://doi.org/10.18653/v1/D19-1461
>
> Howcroft, D.M. & Rieser, V. (2021). What happens if you treat ordinal ratings as interval data? Human evaluations in NLP are even more under-powered than you think. EMNLP 2021. http://doi.org/10.18653/v1/2021.emnlp-main.703
>
> Ju, D., Xu, J., Boureau, Y.-L., & Weston, J. (2022). Learning from data in the mixed adversarial non-adversarial case: Finding the helpers and ignoring the trolls. https://doi.org/10.48550/ARXIV.2208.03295

---

### Official Review · Reviewer_U2Js · 2023-08-17

**Soundness:** 2

**Excitement:**

2: Mediocre: This paper makes marginal contributions (vs non-contemporaneous work), so I would rather not see it in the conference.

**Paper Topic And Main Contributions:**

The paper offers a perspective on addressing the challenges posed by trolls in chatbot robust learning. While the approach is promising, there are inherent limitations, especially concerning data requirements and potential privacy issues.

**Reasons To Accept:**

1. The paper establishes a connection between chatbot robust learning and the Automated Essay Scoring (AES) literature, which is innovative.
2. The paper delves into the realm of robust learning, discussing the threat posed by trolls and the potential solutions. The paper introduces the LCA model for dialogue safety. The model provides a way to determine the probability distribution of observed labels for each user, helping in identifying trolls and helpers.


**Reasons To Reject:**

1.  The AES-like solution proposed has inherently higher data requirements, which might be burdensome for users.
2.  The method might raise logistical and privacy issues, especially when example utterances come from users' own conversations with the chatbot.
3.  The innovation of the algorithm is trivial. The writing of the paper is not very clear. The experiments part is suggested to be organized better.


**Reproducibility:**

3: Could reproduce the results with some difficulty. The settings of parameters are underspecified or subjectively determined; the training/evaluation data are not widely available.

**Reviewer Confidence:**

3: Pretty sure, but there's a chance I missed something. Although I have a good feel for this area in general, I did not carefully check the paper's details, e.g., the math, experimental design, or novelty.

---

> ### Author Rebuttal · Authors · 2023-08-29
>
> Thank you for this review.
>
> # Recap of the experiments
>
> The Reviewer said:
>
> > The writing of the paper is not very clear. The experiments part is suggested to be organized better.
>
> I'll take this opportunity to recap the experiments, to give myself and the Reviewer another chance at a shared understanding.
>
> When faced with trolls, we want to do two things.
> First, we want to de-troll the observed labels.
> Second, we want to carry out supervised learning as normal, with the de-trolled data (Lines 42--44).
>
> To evaluate two imputation methods, MV (majority vote) and LCA (latent class analysis), Experiment 1 did as follows in each run (Lines 177--220).
>
> 1. From Meta's single-turn safety dataset, sample utterances with known gold labels. The result is the training set and the test set.
> 2. Assign training set utterances to users, who label it safe vs. unsafe.
> Some users are trolls, while others are helpers. The scenarios (Lines 203--210) controlled how prevalent trolls were and how they behaved.
> The result is the inter-rater matrix $Z$ (similar to Table 1, above Line 42).
> 3. For each utterance (i.e. row of $Z$), impute (MV or LCA) a de-trolled label.
> 4. Using the de-trolled data, do supervised learning as normal.
> Metric is test set accuracy.
>
> But there's a limitation to using test set accuracy.
> What is of primary interest is the immediate quality of the imputation procedure,
> but the metric conflates that with the quality of the fine-tuned classifier (Line 283).
> One might contend that good results for my proposed AES (automated essay scoring)-like solution are just because it was carried by a good pretrained model or an easy discrimination task.
>
> Thus, I did Experiment 2, where I evaluated the accuracy of the imputation itself (not of the classifier) in the training set.
> Steps 1--3 above are the same, but Step 4 is replaced by a comparing gold vs. imputed labels on the training set (Line 233).
> Note that "training set accuracy" is a regrettable name, as the imputation procedure (MV or LCA) is unsupervised---there is no worry that error is understated.
>
> # Is it trivial?
>
> The Reviewer said:
>
> > The innovation of the algorithm is trivial.
>
> I plead the Reviewer give an explanation of this comment, preferably backed by a citation.
>
> AES (automated essay scoring) has long history (Attali & Burstein 2006), yet the reference (Uto & Okano 2020) inspiring the present paper's methodology is recent.
> I would hesitate to call it "trivial" in the AES context, and I would likewise hesitate to call "trivial" its application to trolls.
>
> I must emphasize that the insight offered by the present paper---that even trolling behavior can be leveraged to benefit classification performance---is counterintuitive. Keep in mind that Ju et al's (2022) CV (cross-validation)-based method simulates tapping into wisdom of the crowd to de-troll data. My work dares to ask, what if the consensus itself is tainted? One could be forgiven for supposing that a diligent-troll majority would be a hopeless situation---a checkmate, then our chatbot devolves into Tay. And in fact, this intuition is supported in the results for MV, as seen in Figure 2 (above Line 242). But it turns out, with LCA, there may still be hope. So indeed, we can learn to love diligent trolls. I plead the Reviewer consider, would you have easily thought so after seeing Ju et al (2022) but before seeing the present paper?
>
> Clearly, Meta AI (Dinan et al 2019; Ju et al 2020) has a stake in their counter-trolling efforts working well.
> As a lone researcher, I humbly offer the aforementioned insight as my piece of the puzzle.
> If this insight is underappreciated in their recent work (Ju e tal 2022), wouldn't NLP researchers be interested to hear that?
>
> # Other comments
>
> The Reviewer said:
>
> > The AES-like solution proposed has inherently higher data requirements, which might be burdensome for users.
>
> The higher data requirements will not stop everyone worried about trolls.
> At any rate, I must emphasize that having the multiple-raters-per-utterance redundancy allows us to tap into a more concrete (or real) consensus, avoiding the simulated consensus from CV---and doing so allows us to evaluate the imputation procedure itself, disentagled from the power of the pretrained models in the background.
> With the merit of LCA itself established in the present paper, future work can remedy this limitation.
> I am personally enthusiastic about the prospect of integrating CV with the AES-like solution.
>
> The Reviewer also said:
>
> > The method might raise logistical and privacy issues, especially when example utterances come from users' own conversations with the chatbot.
>
> Anonymizing data is commonplace in NLP.
> Users can evaluate how safe or unsafe utterances are even after they are scrubbed of personal information.
>
> ----
>
> **References**
>
> Attali, Y. & Burstein, J. (2006). Automated essay scoring with e-rater v.2. The Journal of Technology, Learning, and Assessment, 4 (3), 1--31.
>
> Dinan, E., Humeau, S., Chintagunta, B., & Weston, J. (2019). Build it break it fix it for dialogue safety: Robustness from adversarial human attack. EMNLP 2019. https://doi.org/10.18653/v1/D19-1461
>
> Ju, D., Xu, J., Boureau, Y.-L., & Weston, J. (2022). Learning from data in the mixed adversarial non-adversarial case: Finding the helpers and ignoring the trolls. https://doi.org/10.48550/ARXIV.2208.03295
>
> Uto, M. & Okano, M. (2020). Robust neural automated essay scoring using item response theory. AIED 2020. https://doi.org/10.1007/978-3-030-52237-7_44

---

### Official Review · Reviewer_5UUg · 2023-08-22

**Soundness:** 4

**Excitement:**

3: Ambivalent: It has merits (e.g., it reports state-of-the-art results, the idea is nice), but there are key weaknesses (e.g., it describes incremental work), and it can significantly benefit from another round of revision. However, I won't object to accepting it if my co-reviewers champion it.

**Paper Topic And Main Contributions:**

Inspired by the automated essay scoring (AES) method, this paper proposes to perform latent class analysis (LCA) to infer correct labels in the scenario of de-trolling the training data of utterances generated by chatbots. The presented work is claimed to address the two limitations of the cross-validation (CV) method: being costly, and the risk of being overwhelmed by trolls. The proposed solution first generates the inter-rater matrix Z, and then fits a rater effects model to Z, so that it could infer the label of each utterance suitable for training. Experimental results show the advantage of LCA over majority vote (MV) when the corruption action is designed to be diligent.

**Questions For The Authors:**

A. Line 194: What is rationale behind the setting of max-train-steps? Have you tried other values and did you see any difference?

B. Why did you design two experiments? What is your conclusion when you compare the results of these two experiments?

**Reasons To Accept:**

- The paper proposes a simple and straightforward method to address the problem of trolling in assessing the safety of chatbot utterances.
- Compared with existing methods, LCA is cost-friendly and does not require GPU resources.

**Reasons To Reject:**

- As the experimental results show, the LCA method only outperformed the baseline when the corruption action is "diligent", which is not very realistic in practice---it seems a lot of trolls just give random answers, corresponding to the "lazy" corruption action.
- As the author mentioned, the proposed method has its limitations: it has high data requirements, and it just simply assumes the safe class is the majority of the utterances.

**Reproducibility:**

5: Could easily reproduce the results.

**Reviewer Confidence:**

4: Quite sure. I tried to check the important points carefully. It's unlikely, though conceivable, that I missed something that should affect my ratings.

---

> ### Author Rebuttal · Authors · 2023-08-28
>
> It is clear that Reviewer understood the present paper, based on their summary of it as well as their questions.
> Addressing their questions will certainly improve the present paper.
> Thank you.
>
> # Two experiments?
>
> Let me tackle Question B first.
> The Reviewer said:
>
> > B. Why did you design two experiments? What is your conclusion when you compare the results of these two experiments?
>
> The two experiments have the same scenarios, but they use different metrics.
> Let me explain each one.
>
> **Experiment 1: downstream accuracy.**
> This was done to have a metric in line with Ju et al (2022).
> To evaluate their proposed de-trolling method, what Ju et al (2022) did was: apply it on a trolled training set, fine-tune a classifier on the resulting de-trolled training set, then compute accuracy on a test set.
> But there's an issue---test-set accuracy conflates the performance of the de-trolling method itself with the subsequent fine-tuning (Line 283).
> One might contend that, for instance, good results for my proposed AES (automated essay scoring)-like solution are just because it was carried by a good pretrained model or an easy discrimination task (say, the unsafe utterances are obvious insults).
>
> **Experiment 2: imputation accuracy.**
> For a more direct evaluation, I simply computed the accuracy of the imputed labels in the training set (Line 233).
> Consequently, evaluation was disentangled from fine-tuning.
> Note that the imputation methods compared, LCA (latent class analysis) and MV (majority vote), are both unsupervised.
> Calling the metric "training set accuracy" (Line 233) is regrettable, as there is really no concern that the evaluation understates error.
>
> Another asset of Experiment 2 is that because it avoids LLM fine-tuning, it is cheap. Evaluation is then more precise, done with many many more runs (Line 291).
>
> Altogether now, what can we conclude?
> From two separate metrics, the conclusion is the same:
> Under the AES-like solution, diligent trolls are not a threat, even if they are a majority.
> To see this, notice that in the diligent-troll panels (first two columns) of Figure 2 (above Line 242), the points are far above the horizontal line (marks 70% accuracy, if you predict the majority class all the time) and far above the identity line (marks equal accuracy between LCA and MV).
>
> Why do Experiment 1 at all?
> While it had clear limitations, I imagine that if I had not included it, reviewers would have penalized the present paper, due to the metric's precedence in Ju et al (2022).
> I will be perfectly happy to relegate Experiment 1 to the Appendix, or remove it entirely, if the Reviewer recommends doing so.
>
> # Max train steps?
>
> Now for Question A.
> The Reviewer said:
>
> > A. Line 194: What is rationale behind the setting of max-train-steps? Have you tried other values and did you see any difference?
>
> The parameter `max-train-steps`, which is for fine-tuning with the imputed labels, was set to 400 (Line 195) simply due to limited GPU resources.
> No, I have not tried other values. (Sorry, new experiments ran addressed other Reviewers, and there was not much time.)
> But I do not think of it as a huge issue.
> Note that the two methods compared, LCA and MV, use the same setting, so neither has an unfair advantage in Experiment 1.
> If I were to speculate, more training steps (short of overfitting or catastrophic forgetting) might benefit test-set accuracy in situations where training accuracy was good but might harm test-set accuracy where training accuracy was bad.
> In Experiment 2, fine-tuning was avoided, so the concern of "not enough training was done" (or even "there was too much") disappears.
>
> # The merit of studying diligent trolls
>
> The Reviewer said:
>
> > the LCA method only outperformed the baseline when the corruption action is "diligent", which is not very realistic in practice---it seems a lot of trolls just give random answers, corresponding to the "lazy" corruption action.
>
> A citation would help to convince me that diligent trolling is rare enough to be ignored.
> On my end, I humbly offer that nonrandom adversarial labels are a perfectly respectable topic in crowdsourcing contexts, as in Wang et al (2014).
> Also, the trolling behaviors considered in Ju et al (2022) are not all random.
>
> I must emphasize that the insight offered by the present paper---that even trolling behavior can be leveraged to benefit classification performance---is counterintuitive.
> Keep in mind that Ju et al's (2022) CV (cross-validation)-based method simulates tapping into wisdom of the crowd to de-troll data.
> My work dares to ask, what if the consensus itself is tainted?
> One could be forgiven for supposing that a diligent-troll majority would be a hopeless situation---a checkmate, then our chatbot devolves into Tay.
> And in fact, this intuition is supported in the results for MV, as seen in Figure 2 (above Line 242).
> But it turns out, with LCA, there may still be hope.
> So indeed, we can learn to love diligent trolls.
> I plead the Reviewer consider, would you have easily thought so after seeing Ju et al (2022) but before seeing the present paper?
>
> # Data requirements, safe-as-majority
>
> The Reviewer said:
>
> > As the author mentioned, the proposed method has its limitations: it has high data requirements, and it just simply assumes the safe class is the majority of the utterances.
>
> **On the data requirements.**
> Perhaps the Reviewer will agree that resorting to "more data" is instinctive to many researchers---and to them, the present paper's results are of interest.
> My results show that "more data" is not enough---MV uses the same data as LCA, but MV is more easily misled by a trolled consensus.
> There is a wise way of using more data, and there is an unwise one.
>
> **On the safe-is-majority assumption.**
> Correct inferences must come from somewhere: either one leverages large representative data (so assumptions are unnecessary); or one leverages strong correct assumptions (so even less-than-large data will work).
> Robust continual learning amid trolls is precisely a context where we refuse to put all our trust in data, so some assumptions are necessary.
> In Ju et al (2022), the assumption was that most users act in good faith.
> In the present paper, the assumption is that the chatbot has a base level of decency (Line 161).
> To parties who have a system ready for deployment but are wary of trolls (Line 27), I suspect that the latter assumption is more palatable than the former.
>
> All that said, I look forward to future work remedying both limitations.
> Keep in mind that I have made an effort to evaluate the AES-like solution in a way that disentagles it from fine-tuning LLMs.
> With the basic solution established, the next step might be to reintegrate LLMs to avoid assumptions on which class is more prevalent (Line 162).
>
> ----
>
> **References**
>
> Ju, D., Xu, J., Boureau, Y.-L., & Weston, J. (2022). Learning from data in the mixed adversarial non-adversarial case: Finding the helpers and ignoring the trolls. https://doi.org/10.48550/ARXIV.2208.03295
>
> Wang, G., Wang, T., Zheng, H., & Zhao, B.Y. (2014). Man vs. machine: Practical adversarial detection of malicious crowdsourcing workers. In 23rd USENIX security symposium, USENIX Association, CA. https://www.usenix.org/system/files/conference/usenixsecurity14/sec14-paper-wang-gang.pdf

---

### Meta-Review · Area_Chair_XTV9 · 2023-09-25

**Recommendation:** 3

**Metareview:**

This paper aims to recover the real harmfulness of chatbot conversations even when some users (a.k.a. trolls) intentionally provide incorrect feedback. The proposed method, based on LCA, is significantly more efficient than common cross-validation practices (but also has higher data requirements) and more robust to scenarios where trolls are the majority, provided that they are consistent in their behaviour. Hence, an interesting aspect of this paper is that it presents a somewhat counter-intuitive result. However, the paper also suffers from some weaknesses: the test data size is very limited, which somewhat puts into question the validity of the conclusions. Most crucially, the effectiveness of the method relies on somewhat strong assumptions: namely, 1) that trolls are "diligent", i.e. consistent and 2) that the safe class is in the majority of the utterances. It remains unclear how realistic these assumptions are, without any quantitative evidence from real-world scenarios in support. For these reasons, this paper might be considered for Findings.

---

### Decision · Program_Chairs · 2023-10-07

**Decision:**

Accept-Findings

**Comment:**

This paper aims to recover the real harmfulness of chatbot conversations even when some users (a.k.a. trolls) intentionally provide incorrect feedback. The proposed method, based on LCA, is significantly more efficient than common cross-validation practices (but also has higher data requirements) and more robust to scenarios where trolls are the majority, provided that they are consistent in their behaviour. Hence, an interesting aspect of this paper is that it presents a somewhat counter-intuitive result. However, the paper also suffers from some weaknesses: the test data size is very limited, which somewhat puts into question the validity of the conclusions. Most crucially, the effectiveness of the method relies on somewhat strong assumptions: namely, 1) that trolls are "diligent", i.e. consistent and 2) that the safe class is in the majority of the utterances. It remains unclear how realistic these assumptions are, without any quantitative evidence from real-world scenarios in support. For these reasons, this paper might be considered for Findings.